# A Simple but Effective Approach for Unsupervised Few-Shot Graph Classification

## ABSTRACT

Graphs, as a fundamental data structure, have proven efficacy in modeling complex relationships between objects and are therefore found in wide web applications. Graph classification is an essential task in graph data analysis, which can effectively assist in extracting information and mining content from the web. Recently, few-shot graph classification, a more realistic and challenging task, has garnered great research interest. Existing few-shot graph classification models are all *supervised*, assuming abundant labeled data in base classes for meta-training. However, sufficient annotation is often challenging to obtain in practice due to high costs or demand for expertise. Moreover, they commonly adopt complicated meta-learning algorithms via episodic training to transfer prior knowledge from base classes. To break free from these constraints, in this paper, we propose a simple yet effective approach named SMART for unsupervised few-shot graph classification without using any labeled data. SMART employs transfer learning philosophy instead of the previously prevailing meta-learning paradigm, avoiding the need for sophisticated meta-learning algorithms. Additionally, we adopt a novel mixup strategy to augment the original graph data and leverage unsupervised pretraining on these data to obtain the expressive graph encoder. We also utilize the prompt tuning technique to alleviate the overfitting and low fine-tuning efficiency caused by the limited support samples of novel classes. Extensive experimental results demonstrate the superiority of our proposed approach, significantly surpassing even leading supervised few-shot graph classification models. Our anonymous code is available here.

## CCS CONCEPTS

• **Information systems** → **Data mining**; • **Computing methodologies** → **Neural networks**.

## KEYWORDS

Few-shot learning, Graph neural networks, Unsupervised learning

**ACM Reference Format:**
Anonymous Author(s). 2018. A Simple but Effective Approach for Unsupervised Few-Shot Graph Classification. In *Proceedings of Make sure to enter the correct conference title from your rights confirmation emai (Conference acronym 'XX)*. ACM, New York, NY, USA, 11 pages. https://doi.org/XXXXXXX.XXXXXXX

## 1 INTRODUCTION

A wealth of data on the web can be presented by graphs, which has fostered the flourishing development of a wide range of Web applications [43]. As a fundamental task in graph data mining, graph classification aims to accurately predict the labels of given graphs. Many real-world scenarios can be formulated as this task, such as molecular property predictions [66], social network analysis [59], and online article classification [34]. Recently, effectively learning graph-level representations via graph neural networks (GNNs) and making predictions based on the learned representations has become the dominant paradigm in this field [15, 56]. However, the remarkable performance achieved by these GNN-based models relies on numerous labeled instances. When faced with label-scarcity scenarios, the trained models suffer from severe overfitting and significant performance degradation [9, 35]. Hence, the problem of few-shot graph classification, where only limited support data is available to label query data, has attracted continuous attention from researchers. Several efforts have been made to address this problem and have achieved desired performance [5, 37, 58].

Despite their fruitful success, they all assume the existence of *sufficient labeled graphs* in the base classes for meta-training. However, in the real-world scenario, this assumption is easily shattered as the number of labeled graphs in the base classes is limited or, in some cases, there are no available labeled graphs due to the impracticality, demanding domain knowledge or high cost associated with the annotation process [19, 53]. For example, conducting drug testing in the field of biomedical research requires expensive in-vivo experiments and labor-intensive wet experiments to label drug and protein graphs [8, 19]. In light of this situation, a natural question arises: *Can we break free from reliance on label information while extracting transferable knowledge from these unlabeled graphs that can help adapt to novel classes?* One possible approach is to employ self-supervised learning. However, a formidable challenge is that the limited amount of data can greatly restrict the performance of self-supervised learning [32]. Graph datasets typically have a few base class samples, such as the ENZYMES dataset [44] with only 207 base class samples, which is much fewer than those in image and text datasets. For example, the miniImageNet dataset [57] commonly used for few-shot image classification has 48,000 base class samples. Directly applying self-supervised learning on the base data may result in suboptimal performance. Therefore, it is necessary to expand the diversity of base data to fully utilize the advantage of self-supervised learning.

Moreover, previous studies are developed under complicated meta-learning algorithms by the episodic training manner. Concretely, it samples numerous meta-training tasks (or episodes) from a specific task distribution in the base classes to simulate the real test environment, which aims to quickly transfer prior knowledge from the base classes to the novel ones. Nevertheless, this paradigm not only requires intricate manual model design, but also is prone

to overfitting due to the defined complex hypothesis space [55]. Hence, is there an elegant approach that can ensure model performance while avoiding sophisticated meta-learning algorithms? Meanwhile, during the meta-testing stage, existing models only utilizes limited support set of novel classes to fine-tune the entire model with numerous parameters trained in the meta-training stage, which inevitably leads to overfitting and low fine-tuning efficiency. Therefore, how to alleviate model overfitting and further improve fine-tuning efficiency with limited data is crucial for few-shot graph classification, yet remains unexplored.

To address these limitations mentioned above, we propose a **SiM**ple but effective **A**pproach for unsupe**R**vised few-sho**T** graph classification, coined **SMART**. Specifically, we use a simple transfer learning paradigm to replace the complicated meta-learning paradigm. It necessitates merely pretraining a graph encoder via self-supervised learning on a single task amalgamated from the unlabeled graphs during pretraining, instead of employing meticulously crafted episodic learning-based algorithms. Contrastive learning, as a kind of typical self-supervised techniques, can spontaneously discover supervisory signals from the data itself to learn distinctive representations without explicit labels. Thus, we employ it as the pretraining method to obtain a powerful graph encoder. To increase the diversity of the original data, a straightforward method is to apply mixup for linear interpolation. However, unlike regular Euclidean data such as images and texts, graph-structured data exhibits interdependency and irregularity, meaning two randomly sampled graphs generally have different numbers of nodes and topological structures. It is infeasible that directly applying classical mixup on graphs. To this end, we employ a novel mixup strategy to augment both the unlabeled graphs during pretraining and a few labeled graphs of support set during fine-tuning. Given a pair of graphs, we first compute a transition matrix based on the feature similarity of nodes. Then, we apply this matrix to transform the feature and topology of one graph to align with those of the other. By doing so, we can perform interpolation on any given pair of graphs to synthesize new graphs for training.

Moreover, to mitigate the risk of model overfitting and enhance the efficiency of model fine-tuning, inspired by the huge success of prompt-tuning in the natural language processing, we attempt to introduce this technique to shift its powerful capabilities to our targeted problem. Prompt-tuning typically prepends language prompts in the input text to better guide the language model to understand new tasks. However, given the vast differences between graphs and texts, it is infeasible to design discrete language prompts for target tasks. Recently, a series of studies utilize continuous vectors as soft prompts in the embedding space and have achieved promising performance [22, 28, 52]. Therefore, instead of creating discrete prompts in the original graph data space, we add the trainable vectors as graph-specific features into graphs during fine-tuning, which facilitates extracting extensive knowledge from the pretrained graph encoder. Particularly, we further provide in-depth analysis on the rationale of how prompt-tuning work in our task. In a nutshell, our main contributions are summarized as follows.

• We propose a simple but effective approach named SMART for few-shot graph classification tasks without using any labels. To the best of our knowledge, this is the first work to specifically solve this task by the unsupervised manner.

• We attempt to replace the meta-learning paradigm with transfer learning, and employ a graph-specific mixup strategy to augment the graph data while also utilizing prompt-tuning techniques to further boost model performance.

• We conduct extensive experiments on several datasets. The results demonstrate the superiority of the proposed approach compared to other models, which even outperforms leading supervised models by a large margin.

## 2 RELATED WORK

We briefly review related work from the following four aspects, *i.e.*, graph classification, few-shot learning on graphs, graph contrastive learning, and graph prompt-tuning.

**Graph Classification.** Current graph classification models can be mainly categorized into two classes [58]. The first class is graph kernel methods, which used to be the dominant technique for graph classification [26]. These methods typically leverage kernel functions to measure the similarity between two given graphs, and then easily classify graphs using the SVM classifier. The representative graph kernels include Shortest Path (SP) [3], Graphlet [46], and Weisfeiler-Lehman (WL) [45]. The second class encompasses GNN-based models [24, 33, 56], which leverage GNNs to iteratively aggregate information from neighboring nodes with specific aggregation mechanisms to learn informative latent embeddings of target nodes. A readout function is then applied to aggregate the updated node features to obtain the whole graph embedding for classification. The readout function can be designed as a simple permutation invariant function such as summation or averaging, or a more complicated graph-level pooling function, such as differentiable pooling [63] and self-attention pooling [27]. However, GNN-based models typically rely on a significant amount of labeled graph supervision to achieve optimal performance. When applied directly to scenarios with limited labeled data, *i.e.*, few-shot scenarios, these models may experience severe overfitting, leading to a significant decline in performance.

**Few-shot Learning on Graphs.** The goal of few-shot learning is to rapidly adapt to new tasks with only a few labeled data by leveraging meta-knowledge learned from the abundant training data of base classes [11, 47]. Most previous models address this problem by employing a meta-learning paradigm, which have achieved remarkable success in few-shot scenarios such as few-shot image classification and few-shot text classification. Some classical methods include MAML [11], prototypical networks (PN) [47], and relation networks (RN) [51]. Recently, there has been a growing interest in few-shot learning on graph-structured data. Several studies have focused on integrating GNNs with meta-learning algorithms for various downstream tasks, including few-shot node classification [9, 20, 35, 53, 67], few-shot graph classification [5, 8, 37, 58], and few-shot link prediction [1, 6, 62]. In our focused task, AS-MAML [37] directly integrates MAML with GNNs, enabling efficient adaptation and capturing of substructures in previously unseen graphs for classification. FAITH [58] constructs a hierarchical task graph to model task correlations and employs a loss-based strategy for task sampling to facilitate classification. These specialized models for few-shot graph classification are heavily dependent on labeled data from base classes, which restricts their practical applications.

**Graph Contrastive Learning.** The core concept of contrastive learning is to maximize agreement between embeddings of positive sample pairs while minimizing agreement between negative sample pairs in the embedding space. It has achieved great success in the field of computer vision [7, 17]. Subsequently, contrastive learning has been introduced to the graph domain and has proven to be effective for node representation learning. Some efforts have been made to demonstrate the great potential of contrastive learning on graphs. For example, InfoGraph [48] maximizes mutual information between graph-level representations and substructures at different granularities, enabling the graph-level representations to encode shared information from various substructures. MVGRL [16] trains the graph encoder by maximizing mutual information between representations obtained from different structural views of the graph. Although these methods can obtain decent node embeddings when applied directly to few-shot scenarios, the scarcity of support samples during meta-testing leads to overfitting of the classifier, resulting in suboptimal performance.

**Graph Prompt-tuning.** The concept of prompt-tuning in natural language processing is to facilitate adaptation of pretrained language models to various downstream tasks while reducing the number of parameters used. This is achieved by incorporating learnable soft prompts [29]. Current studies follow the similar paradigm by freezing the parameters of pretrained language models and introducing a few trainable prompt parameters to bridge the gap between downstream tasks and pretraining objectives [13, 18, 30]. Prompt-tuning approaches have achieved tremendous success in natural language processing and have recently been introduced to graph learning as well [10, 36, 50]. Two representative works are GPPT [49] and GraphPrompt [36], which both use link prediction for pretraining and introduce virtual prototype nodes with learnable links into the input graphs to make downstream tasks more compatible with link prediction. However, these methods face challenges when it comes to adapting to other pretraining tasks.

## 3 PRELIMINARY

In this section, we first introduce the notations used and then formally define the problem studied and give relevant preliminary knowledge.

Given a graph dataset $\mathcal{D} = \{\mathcal{G}_i\}_{i=1}^{\varkappa}$, we can denote each graph as $\mathcal{G}_i = \{\mathcal{V}_i, \mathcal{E}_i, \mathbf{X}_i, \mathbf{A}_i\}$, where $\mathcal{V}_i, \mathcal{E}_i, \mathbf{X}_i \in \mathbb{R}^{|\mathcal{V}_i| \times d}$, and $\mathbf{A}_i \in \mathbb{R}^{|\mathcal{V}_i| \times |\mathcal{V}_i|}$ are the set of nodes, the set of edges, the $d$-dimensional node feature matrix, and the adjacency matrix of the graph $\mathcal{G}_i$, respectively. Moreover, we denote the set of graph classes as $C$, which can be further divided into two disjoint sets $C_{ba}$ and $C_{no}$. Here, $C_{ba}$ represents the set of graph *base classes* that are available during meta-training. $C_{no}$ represents the set of graph *novel classes* that need to be predicted given a limited number of labeled graphs during meta-testing. Specifically, $C_{ba} \cup C_{no} = C$ and $C_{ba} \cap C_{no} = \emptyset$. Generally, $C_{no}$ is invisible during meta-training. Previous models all assume that there exists abundant labeled data per class in the base classes for training. Formally, we can define the supervised few-shot graph classification problem as follows:

DEFINITION 1. **Supervised few-shot graph classification:** *Given a graph dataset $\mathcal{D} = \{\mathcal{G}_i\}_{i=1}^{\varkappa}$ and a meta-test task $\mathcal{T} = \{\mathcal{S}, \mathcal{Q}\}$ sampled from $\mathcal{Y}_{C_{no}}$, the goal is to develop a learning model trained on the base **labeled graphs** $\mathcal{B} = (\mathcal{G}_{C_{ba}}, \mathcal{Y}_{C_{ba}})$ that can accurately predict labels for $(\mathcal{G}_q \in \mathcal{G}_{C_{no}})$ (i.e., query set $\mathcal{Q}$) after fine-tuning limited labeled graphs $(\mathcal{G}_s, \mathcal{Y}_s) \in (\mathcal{G}_{C_{no}}, \mathcal{Y}_{C_{no}})$ (i.e., support set $\mathcal{S}$).*

Considering that the labels of base classes are inaccessible in real-world scenarios, we generalize the above definition to unsupervised few-shot graph classification as follows:

DEFINITION 2. **Unsupervised few-shot graph classification:** *Given a graph dataset $\mathcal{D} = \{\mathcal{G}_i\}_{i=1}^{\varkappa}$ and a meta-test task $\mathcal{T} = \{\mathcal{S}, \mathcal{Q}\}$ sampled from $\mathcal{Y}_{C_{no}}$, the goal is to develop a learning model trained on the base **unlabeled graphs** $\mathcal{B} = (\mathcal{G}_{C_{ba}})$ that can accurately predict labels for $(\mathcal{G}_q \in \mathcal{G}_{C_{no}})$ (i.e., query set $\mathcal{Q}$) after fine-tuning limited labeled graphs $(\mathcal{G}_s, \mathcal{Y}_s) \in (\mathcal{G}_{C_{no}}, \mathcal{Y}_{C_{no}})$ (i.e., support set $\mathcal{S}$).*

We can observe that the significant difference between the two definitions lies in whether labels are provided for the base class data. Following the traditional setting in few-shot learning [11], if the support set $\mathcal{S}$ of the test task $\mathcal{T}$ sampled from $\mathcal{Y}_{C_{no}}$ contains $N$ target classes, and each class has $K$ labeled graphs, this is referred to as an $N$-way $K$-shot task. The model is evaluated on the query set $\mathcal{Q}$, which has the same classes as the support set, but with $R$ graphs to be classified per class.

**Graph Neural Networks.** The core concept of GNNs is to iteratively update target nodes by aggregating feature vectors from neighboring nodes through a message passing mechanism [56, 60]. After $k$ iterations of aggregation, the updated representations of the target node can capture structural information within its $k$-order neighbors. Formally, the $k$-layer GNNs can be expressed as:

$$a_v^k = \text{AGG}^k(\{h_u^{k-1} | u \in \mathcal{N}_v\}), \quad h_v^k = \text{COM}^k(h_v^{k-1}, m_v^k) \quad (1)$$

where $a_v^k$ denotes the aggregated message from the set of neighboring nodes $\mathcal{N}_v$. $h_v^k$ is the $k$-layer feature of node $v$ and $h_v^0 = \mathbf{X}$. AGG and COM denote the aggregation and combination functions, which are crucial factors that determine the properties of GNNs. Different choices of the two functions lead to different GNN architectures [15].

For graph-level prediction, we need to apply a readout function on the node representations after the last iteration to obtain the graph representation, which can be formulated as follows.

$$h_{\mathcal{G}} = \text{READOUT}(\{h_v^k | v \in \mathcal{V}\}) \quad (2)$$

where the resulting graph representation $h_{\mathcal{G}}$ is used for the downstream classification task.

## 4 METHOD

In this section, we elaborate on the proposed SMART in detail, which consists of two main components: *graph contrastive pretraining with mixup* and *mixup and prompt-tuning at fine-tuning*. In summary, we obtain a discriminative graph encoder via graph contrastive pretraining. During fine-tuning, we augment the support set for novel classes using mixup while simultaneously improving fine-tuning efficiency via prompt tuning techniques. To better understand our proposed approach, we illustrate its overall architecture in Fig. 1.

**Figure 1: The overall architecture of our proposed SMART.**

## 4.1 Graph Contrastive Pretraining with Mixup

As stated before, previous models all explicitly leverage the labeled data from base classes and are designed with elaborate meta-learning algorithms by the episodic training manner to enable fast adaptation to novel classes under label scarcity scenarios. Here, we adopt a simple transfer learning paradigm, which merges all the unlabeled graph from base classes into a single task. However, we cannot directly perform contrastive learning on the merged dataset due to the limited amount of graphs. Therefore, we propose to augment the base data using mixup techniques. Specifically, we first randomly sample two graphs $\mathcal{G}_1 = \{\mathcal{V}_1, \mathcal{E}_1, \mathbf{X}_1, \mathbf{A}_1\}$ and $\mathcal{G}_2 = \{\mathcal{V}_2, \mathcal{E}_2, \mathbf{X}_2, \mathbf{A}_2\}$, then obtain a transition matrix based on the feature similarity between their nodes. This process can be formulated as:

$$\mathbf{T} = \mathbf{X}_1\mathbf{X}_2^\top \tag{3}$$

where $\mathbf{X}_1 \in \mathbb{R}^{|\mathcal{V}_1| \times d}$ and $\mathbf{X}_2 \in \mathbb{R}^{|\mathcal{V}_2| \times d}$ are the corresponding node feature matrices. $\mathbf{T} \in \mathbb{R}^{|\mathcal{V}_1| \times |\mathcal{V}_2|}$ is the computed transition matrix, where rows represent nodes from graph $\mathcal{G}_1$, and columns represent nodes from graph $\mathcal{G}_2$.

Next, we can use the transition matrix $\mathbf{T}$ to transform the feature and topology of $\mathcal{G}_2$ to align with $\mathcal{G}_1$, which can be expressed as:

$$\tilde{\mathbf{X}}_2 = \mathbf{T}\mathbf{X}_2, \quad \tilde{\mathbf{A}}_2 = \mathbf{T}\mathbf{A}_2\mathbf{T}^\top \tag{4}$$

where $\tilde{\mathbf{X}}_2 \in \mathbb{R}^{|\mathcal{V}_1| \times d}$ and $\tilde{\mathbf{A}}_2 \in \mathbb{R}^{|\mathcal{V}_1| \times |\mathcal{V}_1|}$ are the aligned feature and adjacency matrices. In this way, we acquire the transformed graph $\tilde{\mathcal{G}}_2 = \{\tilde{\mathcal{V}}_2, \tilde{\mathcal{E}}_2, \tilde{\mathbf{X}}_2, \tilde{\mathbf{A}}_2\}$.

We perform linear interpolation on $\mathcal{G}_1$ and $\tilde{\mathcal{G}}_2$ from the perspectives of features and topological structures through mixup operations, defined as:

$$\hat{\mathbf{X}} = \lambda\mathbf{X}_1 + (1-\lambda)\tilde{\mathbf{X}}_2, \quad \hat{\mathbf{A}} = \lambda\mathbf{A}_1 + (1-\lambda)\tilde{\mathbf{A}}_2 \tag{5}$$

where $\hat{\mathbf{X}}$ and $\hat{\mathbf{A}}$ are node feature matrix and adjacency matrix of the generated graph $\hat{\mathcal{G}}$, respectively. $\lambda \in [0, 1]$ is a random variable sampled from the Beta$(\alpha, \alpha)$ distribution parameterized by $\alpha$. By performing Eqs.4 and 5, we can generate sufficient graphs to form an augmented dataset $\hat{\mathcal{B}} = \{\hat{\mathcal{G}}_i\}_{i=1}^m$ to enhance the original base data and increase diversity for pretraining. Note that we do not perform mixup on the labels since labels are inaccessible in this stage.

In the following, we merge the original base data $\mathcal{B}$ with $n$ graphs and augmented base data $\hat{\mathcal{B}}$ with $m$ graphs into one dataset $\mathcal{B}'$ with $(m+n)$ graphs, i.e., $\mathcal{B}' = \mathcal{B} \cup \hat{\mathcal{B}}$, and perform graph contrastive learning on it to learn the intrinsic consistency within the data. Specifically, for each graph in $\mathcal{B}'$, we first perform data augmentation operations, such as node dropping or edge perturbation. In this way, we can obtain $2(m+n)$ graphs in total, with the two correlated augmented views as the positive sample pair, and the remaining $2(m+n-1)$ views as the negative samples.

Next, we feed these graphs into the graph encoder to get the corresponding graph representations. Here, we employ the GIN [60] architecture as the graph encoder, which has been shown to be effective in learning graph-level representations. The specific graph layer of GIN can be described as:

$$h_v^k = \text{MLP}^k((1 + \epsilon^k)h_v^{k-1} + \sum_{u \in \mathcal{N}_v} h_u^{k-1}) \qquad (6)$$

where $h_v^k$ is the node embeddings at $k$ layer of node $v$ and $\epsilon^k$ is the learnable parameter. The final node embeddings are obtained by concatenating the outputs from all layers, i.e., $h_v = h_v^0 || h_v^1 \cdots || h_v^k$. The graph-level representation $h_\mathcal{G}$ is produced by performing summation pooling over all nodes, i.e., $h_\mathcal{G} = \sum_{v \in \mathcal{V}} h_v$. Briefly, we can represent the above graph encoding as $h_\mathcal{G} = f_\theta(\mathcal{G})$, where $\theta$ denotes all the trainable parameters of the graph encoder. Then we use a projection head $g(\cdot)$ to map the graph representation to the hidden space that contrastive loss applied, and conduct $l_2$ normalization, i.e., $z_\mathcal{G} = g(h_\mathcal{G})$ and $z_\mathcal{G} = z_\mathcal{G}/||z_\mathcal{G}||_2$. The concrete graph contrastive loss can be defined as:

$$\mathcal{L}_{cl} = -\frac{1}{2(m+n)} \sum_{k=1}^{2(m+n)} \log \frac{\exp(z_{\mathcal{G}_k} \cdot z_{\mathcal{G}_i}/\tau)}{\sum_{a \in \mathcal{I}} \exp(z_{\mathcal{G}_k} \cdot z_{\mathcal{G}_a}/\tau)} \qquad (7)$$

where $\mathcal{I} = \{1, 2, \cdots, m+n\} \setminus \{k\}$ denotes the set of indices excluding the anchor index $k$ and "$\cdot$" is the inner product. $\tau$ represent a scale temperature parameter. After performing graph contrastive pretraining, we retain only the graph encoder $f_\theta(\cdot)$, which extracts task-agnostic priors, for fine-tuning and discard the remaining components at this stage.

## 4.2 Mixup and Prompt-tuning at Fine-tuning

### 4.2.1 Mixup.
Due to the limited number of support graphs for novel classes in a task during fine-tuning, directly adopting them for linear classification would lead to severe overfitting. To alleviate this issue, we utilize the mixup strategy to enhance the diversity of the support set data. Following the previous stage, we first randomly sample two graphs $(\mathcal{G}_{s_1}, \mathcal{Y}_{s_1})$ and $(\mathcal{G}_{s_2}, \mathcal{Y}_{s_2})$ from the support set $\mathcal{S}$ of task $\mathcal{T}$. Here, $\mathcal{Y}_{s_1}$ and $\mathcal{Y}_{s_2}$ are one-hot class labels of graphs. We then compute the transition matrix $\mathbf{T}_s$ to obtain the aligned $\tilde{\mathcal{G}}_{s_2}$ for $\mathcal{G}_{s_1}$. Finally, we perform linear interpolation on the features and topological structures of the two graphs to generate an augmented graph. The above process can be formulated as follows:

$$\mathbf{T}_s = \mathbf{X}_{s_1}\mathbf{X}_{s_2}^\top, \quad \tilde{\mathbf{X}}_{s_2} = \mathbf{T}_s\mathbf{X}_{s_2}, \quad \tilde{\mathbf{A}}_{s_2} = \mathbf{T}_s\mathbf{A}_{s_2}\mathbf{T}_s^\top,$$
$$\hat{\mathbf{X}}_s = \lambda\mathbf{X}_{s_1} + (1-\lambda)\tilde{\mathbf{X}}_{s_2}, \quad \hat{\mathbf{A}}_s = \lambda\mathbf{A}_{s_1} + (1-\lambda)\tilde{\mathbf{A}}_{s_2} \qquad (8)$$
$$\hat{\mathcal{Y}}_s = \lambda\mathcal{Y}_{s_1} + (1-\lambda)\mathcal{Y}_{s_2}$$

where $\mathcal{G}_{s_1} = \{\mathcal{V}_{s_1}, \mathcal{E}_{s_1}, \mathbf{X}_{s_1}, \mathbf{A}_{s_1}\}$, $\mathcal{G}_{s_2} = \{\mathcal{V}_{s_2}, \mathcal{E}_{s_2}, \mathbf{X}_{s_2}, \mathbf{A}_{s_2}\}$, and $\hat{\mathcal{G}}_s = \{\hat{\mathcal{V}}_s, \hat{\mathcal{E}}_s, \hat{\mathbf{X}}_s, \hat{\mathbf{A}}_s\}$ in which $\hat{\mathcal{G}}_s$ is the generated graph. Note that the significant difference in the fine-tuning phase, is that we explicitly perform mixup operation on the corresponding labels as well. By performing Eq.8, we can synthesize abundant graphs to form an augmented support set $\hat{\mathcal{S}} = \{(\hat{\mathcal{G}}_{s,i}, \hat{\mathcal{Y}}_{s,i})\}_{i=1}^\ell$, and the final support set for fine-tuning can be denoted as $\mathcal{S}' = \mathcal{S} \cup \hat{\mathcal{S}}$. With the above procedure, the adopted mixup extends the data distribution and imposes regularization on the neural network, promoting simplified linear behavior among the training examples and reducing undesirable oscillations in the test data [65].

### 4.2.2 Prompt-tuning.
Although the support graphs for novel classes are enriched through mixup, it is highly inefficient to fine-tune the graph encoder contained numerous parameters and train a linear classifier. Moreover, directly transferring embeddings from the pretrained graph encoder results in suboptimal performance, owing to the inherent discrepancy between the training objectives of the proxy task and that of the downstream few-shot graph classification task. To this end, we add the prompt token to the graph to effectively fine-tune the pretrained graph encoder and customize the pretrained graph embeddings for target graphs. Concretely, we introduce a randomly initialized $d$-dimensional trainable vector as the prompt token, denoted as $P \in \mathbb{R}^d$. Then, we add the prompt vector $P$ to the original features $\mathbf{X}_s$ of the graph from the final support set $\mathcal{S}'$, i.e., $[\mathbf{X}_s + P] \in \mathbb{R}^{|\mathcal{V}_s| \times d}$, and feed them to the pretrained graph encoder. The above procedure can be expressed by:

$$h_{\mathcal{G}_s} = f_\theta(\mathbf{X}_s'), \ \mathbf{X}_s' = \mathbf{X}_s + P = \{\mathbf{X}_{s,1} + P, \mathbf{X}_{s,2} + P, \cdots, \mathbf{X}_{s,|\mathcal{V}_s|} + P\} \qquad (9)$$

where $h_{\mathcal{G}_s}$ is the graph embeddings contextualized by the prompt vector. Note that in this process, we freeze the weights of the pretrained graph encoder and only allow the prompts $P$ to be trainable. In other words, the number of trainable parameters becomes tiny, being just a single vector, which is beneficial for improved fine-tuning efficiency.

*Why prompt-tuning works?* A natural question is, why can such good performance be achieved merely through a single trainable vector, and why is it effective? *From a theoretical perspective*, there has been proven that for any graph prompt function $\Phi(\cdot)$, such as changing node features or adding/removing edges, the following equation holds [10]:

$$f_\theta(\mathbf{A}, \mathbf{X} + P) = f_\theta(\Phi(\mathbf{A}, \mathbf{X})) \qquad (10)$$

This implies that the prompting strategy utilized in this context has the potential to achieve the upper performance bound of any prompting function. If optimizing a specific prompting function $\Phi(\cdot)$ can yield informative graph representations, it follows that directly optimizing the vector $P$ can also achieve the same effect. This intuition arises from the inherent interdependence between the adjacency matrix and the feature matrix in the graph. By appropriately modifying node features, it is possible to influence the structural modifications of the final graph representation. Furthermore, contrary to the commonly acknowledged notion in natural language processing that fine-tuning achieves the theoretical upper bound compared to prompt-tuning [28, 31], this approach demonstrates that conducting prompt-tuning in such a manner can yield superior performance compared to full fine-tuning [10]. *From an empirical perspective*, we find the trained prompt vector tends to focus on the features that are relevant to the graph class. For example, when performing feature selection using Recursive Feature Elimination (RFE) [14] on the protein structure dataset ENZYMES, we find significant overlap between the selected features and the dimensions of the final prompt vector sorted by absolute values, indicating that important features are highlighted.

By performing Eq.9, we can obtain the graph embeddings in the merged support set $\mathcal{S}'$, and train the prompt vectors and a simple

linear classifier using cross-entropy loss, defined as:

$$P^*, \Psi^* = \underset{P, \Psi}{\arg\min} \mathcal{L}_{ce}(\mathcal{S}'; P, \Psi) \quad (11)$$

Finally, we use the resulting prompt vectors $P^*$ and the linear classifier $f_{\Psi^*}$ to predict unlabeled graphs of the query set $Q$ from the task $\mathcal{T}$. The training procedure of SMART can be found in Algorithm 1. Additionally, we derive the *generalization error bound* for the proposed approach by introducing an integral probability metric [40], which depends on the number of graphs in the support set of novel classes, theoretically proving that SMART has good generalization capability. The concrete procedure can be found in **Appendix A.1**.

---

**Algorithm 1** SMART Algorithm

---

**Input:** A graph dataset $\mathcal{D} = \{\mathcal{G}_i\}_{i=1}^{\varkappa}$, Base data $\mathcal{B} = (\mathcal{G}_{C_{ba}})$, A test task $\mathcal{T} = \{\mathcal{S}, \mathcal{Q}\}$.
**Output:** Graph encoder $f_\theta(\cdot)$, linear classifier $f_\Psi(\cdot)$, prompt vector $P$.
  # *Pretraining stage*
1: Obtain augmented data $\hat{\mathcal{B}} = \{\hat{\mathcal{G}}_i\}_{i=1}^{m}$ by mixup with Eqs.3, 4, 5.
2: Feed Merged data $\mathcal{B}' = \mathcal{B} \cup \hat{\mathcal{B}}$ to the graph encoder $f_\theta(\cdot)$ with Eq.6.
3: Train $f_\theta(\cdot)$ by contrastive learning loss with Eq.7.
  # *Fine-tuning stage*
4: Augment the support set $\mathcal{S}$ by mixup with Eq.8.
5: Perform prompt-tuning for the merged support set $\mathcal{S}' = \mathcal{S} \cup \hat{\mathcal{S}}$ with Eq.9.
6: Train a linear classifier $f_\Psi(\cdot)$ using $\mathcal{S}'$.
7: Update $P$ and $\Psi$ by cross-entropy loss with Eq.11.

---

## 5 EXPERIMENT

**Datasets.** To validate the effectiveness of our proposed model, we adopt several datasets widely used for few-shot graph classification [5, 58]. The statistics of these datasets are presented in Table 1, where "Novel" indicates the number of novel classes during the fine-tuning phase. We follow the same train/test class splits as previous studies [5, 58]. The used datasets are described in detail as follows. We also provide the visualization of each dataset in **Appendix** A.2 for clarity.

**Table 1: Statistics of the evaluated datasets.**

| Dataset | # Graphs | # Nodes | # Edges | # Classes | # Novel |
|---------|----------|---------|---------|-----------|---------|
| ENZYMES | 600 | 32.63 | 62.14 | 6 | 2 |
| Letter-high | 2,250 | 4.67 | 4.50 | 15 | 4 |
| Reddit | 1,111 | 391.41 | 456.89 | 11 | 4 |
| TRIANGLES | 2,000 | 20.85 | 35.50 | 10 | 3 |

• **ENZYMES** [44] is a protein tertiary structure dataset composed of enzymes from the BRENDA database, with each class corresponding to a top-level enzyme.
• **Letter-high** [42] contains graphs that represent distorted English letters, where each label denotes the corresponding type of alphabet.

• **Reddit** [61] consists of graphs representing threads, where each node denotes a user, and different graph labels correspond to different types of discussion forums.
• **TRIANGLES** [25] comprises graphs whose classes are determined by the count of triangles/3-cliques present in each graph.

**Baselines.** We compare our proposed method with many **supervised** and **unsupervised** models comprehensively. We mainly select three types of **supervised** models: (I) *classical GNN models*, including **GCN** [24], **GAT** [56], and **GIN** [60]; (II) *classical meta-learning models*, consisting of **PN** [47], **RN** [51], and **MAML** [11]; (III) *supervised few-shot graph classification models*, containing **GSM** [5], **AS-MAML** [37], and **FAITH** [58]. We also select two types of **unsupervised** models: (I) *graph embedding methods*, including **AWE** [21], **Graphlet** [46], **WL** [45], and **Graph2Vec** [41]; (II) *graph contrastive learning methods*, **InfoGraph** [48], **GraphCL** [64], **MVGRL** [16], and **BGRL** [54]. Detailed descriptions of these baselines can be found in **Appendix A.3**. Note that for the *classical GNN models*, we follow the way in [67] to merely modify the dataset split, allowing these methods to be adapted to few-shot settings. For the *graph embedding and graph contrastive learning methods*, we train a Logistic Regression classifier on the learned graph embeddings to perform graph classification. Moreover, we utilize the hyperparameters of these baselines suggested by their original works.

**Implementation Details**. In the pretraining stage, we employ node dropping and attribute masking as graph augmentation techniques and determine suitable ratios by grid search from 0 to 0.4. The number of generated graphs in $\hat{\mathcal{B}}$ is 500, *i.e.*, $m = 500$. We adopt 3-layer GIN with 128 dimensional hidden units as the graph encoder $f_\theta$. In graph contrastive learning, the projection head $g$ is implemented by an MLP layer with a hidden layer and activated by the ReLU function. The temperature parameter $\tau$ in Eq.7 is 0.2. In the fine-tuning stage, the number of generated graphs $\ell$ in $\hat{\mathcal{S}}$ is $20N$, that is, 20 additional graphs are generated by mixup for each class in the support set. We use the Adam [23] method to optimize the whole model. The learning rate and weight decay are 0.001 and 1e-7. An early stopping strategy is utilized during training of the linear classifier $f_\Psi$, where model training is stopped if the training loss does not decrease for five consecutive epochs. All the experiments are conducted in the Python 3.7 and PyTorch 1.13 environment, with a single 24GB NVIDIA GeForce RTX 3090Ti GPU.

**Evaluation Metric.** We adopt accuracy as the metric for evaluating model performance following previous researches [5, 58]. Due to the limited novel categories in used datasets, we set $N = |Novel|$, that is, a testing task contains a support set of $|Novel| \times K$ labeled graphs and a query set of $|Novel| \times R$ unlabeled graphs where $K \in \{5, 10\}$ and $R = 10$. To ensure the stability and fairness of the experiment, we sample 200 testing tasks to evaluate model performance, and repeat the execution 10 times to report the average accuracy and related standard deviation.

## 6 RESULT

**Model Performance.** We show the results of our proposed SMART and other competitive methods on the evaluation datasets in Table 2. According to the results, we observe our proposed method achieve excellent performance on all datasets, achieving the best

**Table 2: Results of different models in various few-shot experimental settings on several datasets. Best: bold. Runner-up: underline.**

| | Model | ENZYMES | | Letter-high | | Reddit | | TRIANGLES | |
|---|---|---|---|---|---|---|---|---|---|
| | | 5-shot | 10-shot | 5-shot | 10-shot | 5-shot | 10-shot | 5-shot | 10-shot |
| Supervised | GCN | 54.30 ± 5.60 | 57.19 ± 4.36 | 60.15 ± 7.62 | 65.67 ± 5.29 | 38.52 ± 3.92 | 41.15 ± 4.22 | 66.39 ± 6.22 | 68.22 ± 3.55 |
| | GAT | 55.40 ± 5.23 | 59.26 ± 5.19 | 66.20 ± 6.67 | 71.25 ± 6.22 | 39.16 ± 5.22 | 42.25 ± 3.55 | 67.50 ± 5.19 | 71.36 ± 3.92 |
| | GIN | 55.73 ± 5.80 | 58.83 ± 5.32 | 65.83 ± 7.17 | 69.16 ± 5.14 | 40.36 ± 4.69 | 43.70 ± 3.98 | 63.80 ± 5.61 | 67.30 ± 4.35 |
| | PN | 53.72 ± 4.37 | 55.79 ± 3.95 | 68.48 ± 3.28 | 72.60 ± 3.01 | 42.31 ± 2.32 | 43.23 ± 2.01 | 69.56 ± 3.97 | 73.12 ± 3.62 |
| | RN | 41.39 ± 4.73 | 43.27 ± 3.49 | 51.14 ± 4.21 | 52.54 ± 4.04 | 34.89 ± 3.76 | 37.76 ± 3.09 | 46.09 ± 3.10 | 49.15 ± 3.49 |
| | MAML | 51.96 ± 7.22 | 53.62 ± 7.19 | 67.49 ± 4.59 | 71.55 ± 5.15 | 31.62 ± 5.11 | 36.49 ± 4.25 | 72.32 ± 3.42 | 74.49 ± 4.62 |
| | GSM | 55.42 ± 5.74 | 60.64 ± 3.84 | 69.91 ± 5.90 | 73.28 ± 3.64 | 41.59 ± 4.12 | 45.67 ± 3.68 | 71.40 ± 4.34 | 75.60 ± 3.67 |
| | AS-MAML | 49.83 ± 1.12 | 52.30 ± 1.43 | 69.44 ± 0.75 | 75.93 ± 0.53 | 36.96 ± 0.74 | 41.47 ± 0.83 | 78.42 ± 0.67 | 80.39 ± 0.56 |
| | FAITH | 57.89 ± 4.65 | 62.16 ± 4.11 | 71.55 ± 3.58 | 76.65 ± 3.26 | 42.71 ± 4.18 | 46.63 ± 4.01 | **79.59 ± 4.05** | **80.79 ± 3.53** |
| Unsupervised | AWE | 43.75 ± 1.85 | 45.58 ± 2.11 | 40.60 ± 3.91 | 42.20 ± 2.87 | 30.24 ± 2.34 | 33.44 ± 2.04 | 39.36 ± 3.85 | 42.58 ± 3.11 |
| | Graphlet | 53.17 ± 5.92 | 55.30 ± 3.78 | 33.76 ± 6.95 | 37.59 ± 4.60 | 33.76 ± 6.94 | 37.59 ± 4.60 | 40.17 ± 3.18 | 43.76 ± 3.09 |
| | WL | 55.78 ± 4.72 | 58.37 ± 3.84 | 65.27 ± 7.67 | 69.39 ± 4.69 | 40.26 ± 5.17 | 42.57 ± 3.69 | 51.25 ± 4.02 | 53.26 ± 2.95 |
| | Graph2Vec | 55.88 ± 4.86 | 58.22 ± 4.30 | 66.12 ± 5.21 | 68.17 ± 4.26 | 27.85 ± 4.21 | 29.97 ± 3.17 | 48.38 ± 3.85 | 50.16 ± 4.15 |
| | InfoGraph | 51.11 ± 2.05 | 56.84 ± 2.75 | 64.83 ± 2.11 | 68.21 ± 2.62 | 39.29 ± 3.66 | 41.32 ± 3.71 | 57.65 ± 1.75 | 64.40 ± 2.50 |
| | GraphCL | 53.69 ± 1.42 | 55.78 ± 1.77 | 69.66 ± 2.42 | 73.04 ± 2.83 | 42.20 ± 4.83 | 45.76 ± 4.19 | 51.13 ± 2.39 | 54.76 ± 1.21 |
| | MVGRL | 48.33 ± 1.65 | 52.63 ± 1.84 | 70.55 ± 5.10 | 72.32 ± 2.43 | 40.15 ± 3.99 | 43.22 ± 3.79 | 48.30 ± 2.29 | 56.89 ± 5.40 |
| | BGRL | 57.78 ± 2.87 | 60.42 ± 2.65 | 51.07 ± 2.69 | 55.00 ± 3.26 | 41.19 ± 3.12 | 44.22 ± 3.99 | 63.78 ± 7.90 | 54.76 ± 1.21 |
| | SMART | **59.80 ± 3.39** | **65.11 ± 2.70** | **74.17 ± 2.75** | **76.89 ± 1.55** | **43.83 ± 2.21** | **47.75 ± 2.77** | 79.39 ± 2.45 | 80.43 ± 2.12 |

results in three of the four datasets and the second-best results in the remaining dataset, which demonstrates its effectiveness for few-shot graph classification tasks. SMART not only outperforms unsupervised models by a large margin, but also surpasses supervised models specifically designed for this task. For example, on the ENZYMES dataset, SMART achieves absolute improvements of 1.91% and 2.95% over FAITH in the 5-shot and 10-shot few-shot scenarios, respectively. We believe that the underlying reason is that the mixup used in the pretraining stage generates a sufficient variety of graphs, which allows the potential of graph contrastive learning to be fully unleashed, resulting in a discriminative graph encoder. Additionally, the mixup used in the fine-tuning stage effectively expands the original data distribution in the support set, resulting in a linear transition of the decision boundary from one class to another, which provides a smoother uncertainty estimation. Moreover, the adopted prompt-tuning strategy further leverages the pretrained graph encoder and improves fine-tuning efficiency with few trainable parameters.

We find that supervised few-shot graph classification models achieve better performance compared to models from other categories. This can be attributed to their tailored designs for few-shot scenarios while explicitly utilizing graph structural information. However, due to the scarcity of support set data, they still suffer from overfitting issues, thus their performance lags far behind our model. Additionally, classic GNN models show unsatisfactory performance, owing to their inability to sufficiently learn graph structural features and category information under the few-shot scenario with insufficient label data, resulting in poor generalization capability. Notably, graph contrastive learning models do not achieve ideal performance, sometimes even underperforming unsupervised graph embedding methods. A possible reason is that the small scale of graph datasets greatly limits the capability of contrastive learning, which reflects the necessity of our data augmentation on base classes.

**Ablation Study.** To validate the effectiveness of the adopted strategies, we conduct comprehensive ablation studies on several designed model variants formed by sequentially adding the utilized techniques over all evaluation datasets. (I) *Raw model*: We perform graph contrastive pretraining on base data to obtain a graph encoder, and then directly utilize the graph embeddings of support set data obtained from the frozen graph encoder to train a linear classifier for evaluating the query set. (II) $+mixup_{pre}$: We use the mixup strategy during pretraining to generate abundant graphs for potentially training a more expressive graph encoder. (III) $+mixup_{test}$: We further leverage the mixup strategy at fine-tuning stage to enrich graph diversity in the support set to alleviate classifier overfitting. (IV) $+pt$: We continue to employ the prompt-tuning strategy at fine-tuning to improve fine-tuning efficiency by bridging the gap between pretraining and downstream tasks.

By analyzing the results shown in Table 3, we can clearly observe that all the adopted strategies are crucial for improving model performance. In particular, the $mixup_{pre}$ strategy used during pretraining brings significant gains to SMART, aligning with our expectation. A reasonable explanation is that it provides more diverse samples, facilitating contrastive learning to learn richer feature representations. Additionally, the $mixup_{test}$ strategy used at fine-tuning stage also leads to considerable improvements for SMART. We attribute this to this operation enriching the limited support set and alleviating the data scarcity issue. Finally, the prompt-tuning

**Table 3: Ablation results on the evaluated datasets.**

| Dataset | ENZYMES | | Letter-high | | Reddit | | TRIANGLES | |
|---|---|---|---|---|---|---|---|---|
| | 5-shot | 10-shot | 5-shot | 10-shot | 5-shot | 10-shot | 5-shot | 10-shot |
| *Raw model* | 53.69 ± 1.42 | 58.78 ± 1.77 | 69.66 ± 2.42 | 73.04 ± 2.83 | 42.20 ± 4.83 | 45.76 ± 3.19 | 73.13 ± 2.39 | 74.76 ± 2.21 |
| $+mixup_{pre}$ | 56.46 ± 2.36 | 62.12 ± 2.55 | 71.39 ± 3.22 | 74.59 ± 1.95 | 42.79 ± 3.29 | 46.79 ± 2.59 | 76.02 ± 2.19 | 77.19 ± 2.35 |
| $+mixup_{test}$ | 58.30 ± 2.25 | 63.29 ± 2.20 | 73.11 ± 2.30 | 75.09 ± 2.58 | 43.02 ± 2.52 | 47.36 ± 2.10 | 78.19 ± 2.10 | 79.06 ± 2.45 |
| $+pt$ | **59.80 ± 3.39** | **65.11 ± 2.70** | **74.17 ± 2.75** | **76.89 ± 1.55** | **43.83 ± 2.21** | **47.75 ± 2.77** | **79.39 ± 2.45** | **80.43 ± 2.12** |

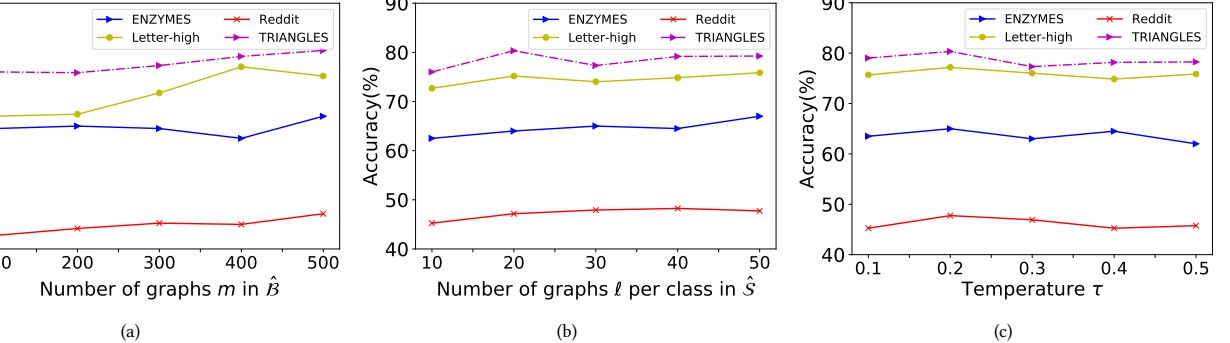

(a)  (b)  (c)

**Figure 2: Evaluation performance with different hyperparameters.**

strategy offers certain improvements, by reducing overfitting and improving fine-tuning efficiency with fewer trainable parameters. **Hyperparameter Sensitivity.** We mainly investigate the effects of three hyperparameters on model performance, with results shown in Fig. 2. All datasets are tested under the $N$-way 10-shot experimental setting. Note that when exploring one hyperparameter, others are fixed to default values. As exhibited in Fig. 2 (a), the model performance steadily increases as $m$ increases, owing to the beneficial diversity for contrastive learning. Fig. 2 (b) shows the model performance also demonstrates increasing trends with larger $\ell$, while the optimal $\ell$ value varies across datasets. Moreover, the satisfactory results can be achieved with relatively small $\tau$. Specifically, as depicted in Fig. 2 (c), we find the peak model performance is generally attained at $\tau = 0.2$.

**Parameter Efficiency.** We compare the number of parameters needed for updating when performing the downstream few-shot graph classification task between our proposed SMART and several representative models. As shown in Table 4, the results demonstrate the tunable parameters in SMART are significantly less than other models, even smaller by orders of magnitude. For example, on the ENZYMES dataset, the tunable parameters of our model are only 0.107% of that in the FAITH model, which sufficiently validates the high tuning efficiency of our model. Since other models require full fine-tuning of parameters, which is expensive in resources and prone to overfitting, while our proposed model alleviates this issue by employing prompt-tuning techniques.

**Table 4: Number of parameters on several datasets. The last row represents the percentage of parameter count of SMART compared to FAITH.**

| Model | ENZYMES | Letter-high | Reddit | TRIANGLES |
|---|---|---|---|---|
| GAT | 165,120 | 153,140 | 224,155 | 166,325 |
| GIN | 120,494 | 119,970 | 187,959 | 120,625 |
| GSM | 294,130 | 294,061 | 361,790 | 29,4456 |
| FAITH | 989,838 | 992,917 | 1,057,297 | 991,505 |
| SMART | 1,035 | 1,545 | 2,064 | 1,293 |
| Ratio | 0.104% | 0.156% | 0.195% | 0.130% |

## 7 CONCLUSION

In this work, we propose a simple but effective approach named SMART for solving few-shot graph classification tasks in an unsupervised manner. Specifically, we adopt a simple transfer learning paradigm to replace the previously complicated meta-learning paradigm. In the pretraining stage, we utilize a modified mixup to enrich data diversity for obtaining a powerful graph encoder. During the fine-tuning stage, we leverage the same modified mixup and prompt-tuning techniques to alleviate the overfitting and inefficient fine-tuning issues caused by limited support set data. Experimental results on multiple datasets demonstrate that our proposed model even surpasses previous competitive supervised models, which sufficiently validates its effectiveness. We hope our work could provide inspirations for future research on unsupervised few-shot graph classification.

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

# A APPENDIX

## A.1 Theoretical Analysis

In this section, we theoretically analyze for the good generalization capability of our proposed approach. The goal of SMART is to rapidly generalize the meta-knowledge obtained from base classes to novel ones, which essentially minimizes the distribution divergence between $\mathcal{G}_{C_{ba}}$ and $\mathcal{G}_{C_{no}}$. Inspired by integral probability metric (IPM) [40] utilized for analyzing various models, we leverage it to derive the upper bound. Formally, the definition of IPM as follows:

$$\gamma_{\mathcal{F}}(\mathbb{P}, \mathbb{Q}) = \sup_{f \in \mathcal{F}, \mathcal{G} \in \mathcal{D}} |\mathbb{E}_{\mathbb{P}} f(\mathcal{G}) - \mathbb{E}_{\mathbb{Q}} f(\mathcal{G})| \quad (12)$$

where $\mathbb{P}$ and $\mathbb{Q}$ are two probability distributions. $\mathbb{E}_{\mathbb{P}}(\cdot)$ denotes the expectation of a variable over a probability distribution $\mathbb{P}$. $\mathcal{F}$ is a class of real-valued bounded measurable functions and $\mathcal{D}$ is the defined data space.

Moreover, we also need to utilize the empirical Rademacher complexity, which is defined as:

$$\mathcal{R}(\mathcal{F}|\mathcal{G}_1, \cdots, \mathcal{G}_U) = \mathbb{E}_{\sigma} \sup_{f \in \mathcal{F}} \frac{1}{U} \left| \sum_{i=1}^{U} \sigma_i f(\mathcal{G}_i) \right| \quad (13)$$

where $\{\sigma_1, \cdots, \sigma_U\}$ are the *i.i.d.* Rademacher random variables.

According to the empirical Rademacher complexity, we have the following lemma about a standard uniform deviation bound [2].

LEMMA 1. **Uniform deviation with empirical Rademacher complexity**: Let $\{\mathcal{G}_1, \cdots, \mathcal{G}_U\}$ be sampled i.i.d. from the distribution $\mathbb{P}$ on $\mathcal{D}$, and let $\mathcal{F}$ denote a class of functions mapping from $\mathcal{D}$ to $[a, b]$. For $\epsilon > 0$, we derive that with a probability at least $(1 - \epsilon)$ over the draw of the sample,

$$\sup_{f \in \mathcal{F}} |\mathbb{E}_{\hat{\mathbb{P}}} f(\mathcal{G}) - \mathbb{E}_{\mathbb{P}} f(\mathcal{G})| \leq 2\mathcal{R}(\mathcal{F}|\mathcal{G}_1, \cdots, \mathcal{G}_U) + 3\sqrt{\frac{(b-a)^2 \log(2/\epsilon)}{2U}} \quad (14)$$

where $\hat{\mathbb{P}}$ denotes the empirical distribution of the sample, and $\mathcal{R}(\mathcal{F}|\mathcal{G}_1, \cdots, \mathcal{G}_U)$ denotes the empirical Rademacher complexity of the function class $\mathcal{F}$.

In our scenario, we denote the empirical data distribution in $\mathcal{G}_{ba}$ as $\hat{\mathbb{P}}$, the empirical data distribution of the enriched support set $\mathcal{S}'$ from the meta-test task as $\hat{\mathbb{Q}}$, and the expected data distribution of the query set $Q$ as $\mathbb{Q}$, respectively. Then, essentially, we aim to

minimize the $\gamma_{\mathcal{F}}(\hat{\mathbb{P}}, \mathbb{Q})$, which satisfies the following equation:

$$\gamma_{\mathcal{F}}(\hat{\mathbb{P}}, \mathbb{Q}) = \sup_{f \in \mathcal{F}} |\mathbb{E}_{\hat{\mathbb{P}}} f(\mathcal{G}) - \mathbb{E}_{\mathbb{Q}} f(\mathcal{G})| \quad (15)$$

We can obtain the upper bound of $\gamma_{\mathcal{F}}(\hat{\mathbb{P}}, \mathbb{Q})$ by introducing the following theorem [4, 39].

THEOREM 1. *For any function $f$ in a class $\mathcal{F}$ and $f(\mathcal{G}) \in [a, b]$, suppose that training and testing data are independent and drawn independent identically distributed (i.i.d.), with a probability at least $(1 - \epsilon)$ over the draws of the training and query samples,*

$$\gamma_{\mathcal{F}}(\hat{\mathbb{P}}, \mathbb{Q}) \leq \gamma_{\mathcal{F}}(\hat{\mathbb{P}}, \hat{\mathbb{Q}}) + 2\mathcal{R}(\mathcal{F}|\mathcal{G}_1, \cdots, \mathcal{G}_U) + 3\sqrt{\frac{(b-a)^2 \log(2/\epsilon)}{2U}} \quad (16)$$

*where $U = \ell + NK$ is the number of enriched support set and $\mathcal{R}(\mathcal{F}|\mathcal{G}_1, \cdots, \mathcal{G}_U)$ is the empirical Rademacher complexity of the function class $\mathcal{F}$ with respect to support graphs.*

*Proof.* We can prove the above theorem as follows:

$$\gamma_{\mathcal{F}}(\hat{\mathbb{P}}, \mathbb{Q}) = \sup_{f \in \mathcal{F}} |\mathbb{E}_{\hat{\mathbb{P}}} f(\mathcal{G}) - \mathbb{E}_{\mathbb{Q}} f(\mathcal{G})|$$

$$= \sup_{f \in \mathcal{F}} |\mathbb{E}_{\hat{\mathbb{P}}} f(\mathcal{G}) + \mathbb{E}_{\hat{\mathbb{Q}}} f(\mathcal{G}) - \mathbb{E}_{\hat{\mathbb{Q}}} f(\mathcal{G}) - \mathbb{E}_{\mathbb{Q}} f(\mathcal{G})|$$

$$\leq_a \sup_{f \in \mathcal{F}} \left[ |\mathbb{E}_{\hat{\mathbb{P}}} f(\mathcal{G}) - \mathbb{E}_{\hat{\mathbb{Q}}} f(\mathcal{G})| + |\mathbb{E}_{\hat{\mathbb{Q}}} f(\mathcal{G}) - \mathbb{E}_{\mathbb{Q}} f(\mathcal{G})| \right]$$

$$\leq_b \sup_{f \in \mathcal{F}} |\mathbb{E}_{\hat{\mathbb{P}}} f(\mathcal{G}) - \mathbb{E}_{\hat{\mathbb{Q}}} f(\mathcal{G})| + \sup_{f \in \mathcal{F}} |\mathbb{E}_{\hat{\mathbb{Q}}} f(\mathcal{G}) - \mathbb{E}_{\mathbb{Q}} f(\mathcal{G})|$$

$$=_c \gamma_{\mathcal{F}}(\hat{\mathbb{P}}, \hat{\mathbb{Q}}) + \sup_{f \in \mathcal{F}} |\mathbb{E}_{\hat{\mathbb{P}}} f(\mathcal{G}) - \mathbb{E}_{\mathbb{P}} f(\mathcal{G})|$$

$$\leq_d \gamma_{\mathcal{F}}(\hat{\mathbb{P}}, \hat{\mathbb{Q}}) + 2\mathcal{R}(\mathcal{F}|\mathcal{G}_1, \cdots, \mathcal{G}_U) + 3\sqrt{\frac{(b-a)^2 \log(2/\epsilon)}{2U}} \quad (17)$$

Inequality (a) becomes valid by utilizing the triangle inequality. Splitting the previous term gives rise to inequality (b). Leveraging Eq.12 leads to the establishment of equality (c). Finally, applying Lemma 1 results in the validity of equality (d).

From the Theorem 1, we can conclude that the generalization error upper bound involves purely empirical quantities, *i.e.*, the empirical IPM and empirical Rademacher complexity. The first entry, *i.e.*, empirical IPM, can be optimized by transferring pretrained graph embeddings during testing. According to [12], the empirical Rademacher complexity is inversely proportional to the number of support graphs $U$, *i.e.*, $\mathcal{R}(\mathcal{F}|\mathcal{G}_1, \cdots, \mathcal{G}_U) \propto \frac{1}{U}$. Assume all the selected methods can optimize to achieve the optimal empirical IPM, but for previous methods, since $U$ is particularly small, the second and third terms become larger, increasing the generalization error bound. While SMART explicitly increases $U$ through mixup, making the second and third terms significantly smaller, even approaching 0, thereby obtaining better generalization capability.

## A.2 Dataset Visualization

We provide visualizations of typical examples from each dataset here, for better clarity, as shown in Fig.3.

## A.3 Baseline Descriptions

**Supervised Models**.
(I) *Classical GNN models.*

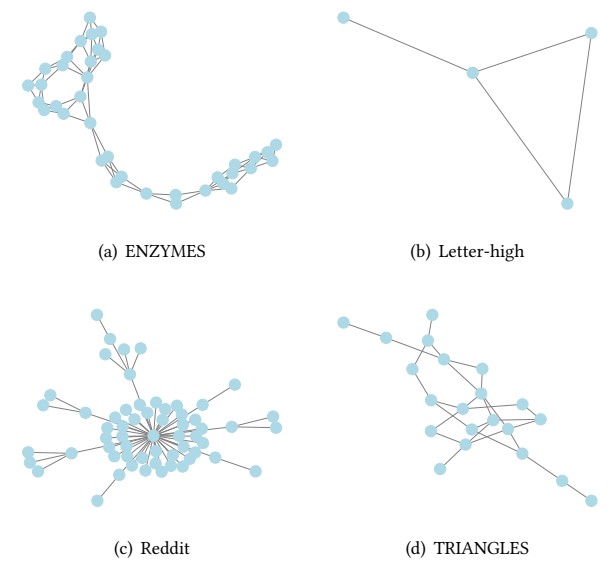

(a) ENZYMES

(b) Letter-high

(c) Reddit

(d) TRIANGLES

**Figure 3: Visualization of a graph contained in each dataset.**

**GCN** [24]: It learns the hidden embeddings of nodes by defining graph convolution operations in the spectral domain.

**GAT** [56]: It assigns different weights to the neighboring nodes of the target node based on their importance in the neighborhood, unlike treating each node equally by previous studies.

**GIN** [60]: It ensures that the aggregation and readout functions of GNNs satisfy the injective property, thereby endowing GNNs with powerful representation comparable to the WL test in distinguishing graph structures.

(II) *Classical meta-learning methods.*

**PN** [47]: It learns the metric function in the embedding space that allows data of the same category to be closer to its prototype while being farther away from prototypes of different categories.

**RN** [51]: It learns to perform deep distance metric learning by computing relation similarity scores between unlabeled data and few labeled data.

**MAML** [11]: It computes the second-order gradients of the model to find optimal initialization parameters, enabling faster adaptation to new categories.

(III) *Supervised few-shot graph classification models.*

**GSM** [5]: It utilizes graph spectral measures to generate a set of super-classes and constructs a corresponding super-graph to model the relationships between classes.

**AS-MAML** [37]: It combines GNNs with MAML for fast adaption on unseen test graphs, and proposes a step controller for improving the robustness of meta-learner.

**FAITH** [58]: It captures task correlations by constructing graphs at different granularities — instance-level, prototype-level, and task-level graphs.

**Unsupervised Models**.

(I) *Graph embeddings models.*

**AWE** [21]: It proposes to leverage the anonymous walks strategy to learn discriminative graph embeddings, which provides characteristic graph traits and accurately reconstructs the network proximity of nodes.

**Graphlet** [46]: It is a well-known graph kernel method that introduces a kernel based on counting the occurrences of fixed-size subgraph patterns to compute the similarity between pairs of graphs.

**WL** [45]: It presents a neighborhood aggregation kernel for graphs with discrete labels based on 1-dimensional WL to solve the graph isomorphism problem.

**Graph2Vec** [41]: It follows the assumption that graphs with similar subgraphs and structures have similar embeddings, and utilizes the Skip-gram model of word2vec [38] to maximize the prediction of the probability of subgraphs existing in the input graph.

(II) *Graph contrastive learning models.*

**InfoGraph** [48]: It learns generalizable graph-structured representations for downstream graph classification tasks by maximizing the mutual information between graph-level representations and representations of different scales of substructures.

**GraphCL** [64]: It introduces various graph augmentations applied to the original graph to incorporate diverse priors, and develops a graph contrastive learning framework for learning unsupervised representation of graph-structed data.

**MVGRL** [16]: It performs graph contrastive learning by extracting first-order neighbors and generalized graph diffusion from the input graph to obtain different structural views and achieve better performance in several downstream tasks.

**BGRL** [54]: It designs a graph contrastive learning architecture that does not require graph augmentation or numerous negative samples, yet still achieves high-quality latent representations.

