# OpenReview forum: "A Simple but Effective Approach for Unsupervised Few-Shot Graph Classification"
_ACM.org/TheWebConf/2024/Conference — TheWebConf24_

### Official Review · Reviewer_Q9kC · 2023-11-23

**Novelty:** 4
**Technical Quality:** 4

**Review:**

This paper proposes a simple and effective approach called SMART for unsupervised few-shot graph classification. This paper replaces the meta-learning paradigm with transfer learning and employs a graph-specific mixup strategy to augment the graph data while also utilizing prompt-tuning techniques to further boost model performance. The paper also provides extensive experimental results on several datasets to demonstrate the superiority of the proposed approach, which even outperforms leading supervised models by a large margin. The code is provided, and the paper is well-written and easy to follow. My concerns are as follows:
1. In the problem setting, the feature vectors of all graphs are assumed to be $d$. Is this a common setting? What if the graphs have different feature sets? The transition matrix $\mathbf{T}$ in Eq. (3) cannot be established. Also, how to ensure the transition matrix $\mathbf{T}$ in Eq. (3) is non-negative since feature matrices $\mathbf{X}_1$ and $\mathbf{X}_2$ might contain negative entries?
2. The time complexity analysis is missing.
3. The mixup approach for data augmentation in graph classification is not new. Please highlight the non-trivial and novel points here.
4. The techniques here seem to be simple and not new. Please provide more theoretical insights on the effectiveness of the proposed solution.

**Questions:**

1. In the problem setting, the feature vectors of all graphs are assumed to be $d$. Is this a common setting? What if the graphs have different feature sets? The transition matrix $\mathbf{T}$ in Eq. (3) cannot be established. Also, how to ensure the transition matrix $\mathbf{T}$ in Eq. (3) is non-negative since feature matrices $\mathbf{X}_1$ and $\mathbf{X}_2$ might contain negative entries?
2. The time complexity analysis is missing.
3. The mixup approach for data augmentation in graph classification is not new. Please highlight the non-trivial and novel points here.
4. The techniques here seem to be simple and not new. Please provide more theoretical insights on the effectiveness of the proposed solution.

**Reviewer Confidence:**

3: The reviewer is confident but not certain that the evaluation is correct

**Scope:**

3: The work is somewhat relevant to the Web and to the track, and is of narrow interest to a sub-community

---

### Official Review · Reviewer_665w · 2023-11-25

**Novelty:** 5
**Technical Quality:** 6

**Review:**

This paper introduces a new graph classification approach SMART targeting unknown classes with limited representation in the training dataset, with an approach for few-shot graph classification in an unsupervised setting. This methodology deviates from traditional meta-learning techniques and instead employs a transfer learning philosophy. The key components of SMART include a new mixup strategy for data augmentation with limited samples from the testing set and the utilization of prompt-tuning techniques to enhance fine-tuning efficiency. These methods are claimed to address the challenges of model overfitting and the limitations posed by scarce labeled data in few-shot learning scenarios.

In the context of the paper "A Simple but Effective Approach for Unsupervised Few-Shot Graph Classification," the term "prompt" as used in prompt-tuning does not refer to a textual prompt in the traditional sense, as seen in natural language processing (NLP). Instead, it relates to adapting the concept of prompt-tuning, commonly used in NLP, to the domain of graph classification.

In NLP, prompt-tuning involves modifying or adding to the input data to guide the model in performing a specific task. For example, if the task is to classify sentiment, a prompt might be a phrase added to the input text that frames the text in a way that the model understands it's supposed to evaluate sentiment.

Translating this concept to graph classification, particularly in an unsupervised few-shot learning context, the "prompt" would be analogous to a mechanism or technique used to prepare or augment graph data in a way that facilitates the classification task. This could involve modifying the graph structure, adding certain features or nodes, or any other form of data manipulation that effectively "prompts" the model to classify graphs correctly, even with limited training data.

In the case of SMART, the paper mentions a mixup strategy for data augmentation and the utilization of prompt-tuning techniques. Here, these strategies act as "prompts" in the sense that they prepare and augment the graph data to make the model more effective at classifying graphs, especially given the challenges of few-shot learning and the absence of large labeled datasets.

**Questions:**

1) What is the broad strategy to perform prompting in the graph scenario? In NLP tasks, it is easy to interpret the prompt using the words in it, since the task is provided in natural language. Is there a simple interpretation aspect for graph prompts?

2) Do we need to learn a separate prompt embedding for each novel class? Is there some shared aspect? What are the similarities to prompt tuning in NLP? What are the key differences?

**Ethics Review Description:**

Not required

**Reviewer Confidence:**

3: The reviewer is confident but not certain that the evaluation is correct

**Scope:**

4: The work is relevant to the Web and to the track, and is of broad interest to the community

---

### Official Review · Reviewer_nwEq · 2023-11-25

**Novelty:** 4
**Technical Quality:** 3

**Review:**

This paper proposes to utilize unsupervised learning, graph mixup, and graph prompt-tuning to address issues in few-shot graph classification tasks.
In pre-training stage, the authors use graph contrastive learning methods to get a graph encoder on data enriched by mixup.
In fine-tuning stage, the authors mitigate the overfitting issue by mixup.
To improve the efficiency of fine-tuning, the author use graph prompt-tuning instead of fine-tuning all the encoder.

# Strengths
S1. Addressing model overfiting is an important topic in the field of few-shot learning.
S2. The method proposed by authors exhibit its effectiveness on few-shot graph classification task.
S3. The trainable parameters in fine-tuning stage are less than ofher mehods.


# Weaknesses
W1. The motivation of this work should be further explained. Is it indeed meaningful to get rid of the label cost at few shots.
W2. The technical novelty of the proposed method is questionable. The core idea is straightforward. The specific techniques used are directly borrowed from other tasks.
W3. The performance of compared methods exhibits high variance. It is unclear whether the performance improvement of the proposed method is significant.
W4. There lacks ablation study of unsupervised methods. I wonder if supervised with mixup and prompt-learning could obtain better performance.

**Questions:**

Q1. Do you conduct experiments on this setting: using unsupervised learning methods train encoders on additional dataset which is larger, unlabeled, and similar to the task to exhibit the superior of unsupervised methods?

Q2. There are some graph mixup methods, such as _G-Mixup: Graph Data Augmentation for Graph Classification_, do you conduct experiment to compare these methods with your mixup methods.

Q3. Why the meta-learning methods get worse performance than classical GNN models on ENZYMES and Reddit datasets.

Q4. The ablation experiment shows that the unsupervised learning methods get worse performance, do you conduct experiments to evaluate using mixup and prompt-tuning on supervised methods.

**Reviewer Confidence:**

3: The reviewer is confident but not certain that the evaluation is correct

**Scope:**

2: The connection to the Web is incidental, e.g., use of Web data or API

---

### Official Review · Reviewer_DT48 · 2023-11-28

**Novelty:** 4
**Technical Quality:** 5

**Review:**

The paper has merits. It proposed an approach named SMART for solving few-shot graph classification tasks in an unsupervised manner. Instead of using the meta-learning paradigm, SMART adopted a transfer learning paradigm. In the pretraining stage, a modified mixup strategy is used to enrich data diversity for obtaining a powerful graph encoder. During the fine-tuning stage, SMART leveraged the same modified mixup and prompt-tuning techniques to alleviate the overfitting and inefficient fine-tuning issues caused by limited support set data. Extensive experiments have been done to prove its efficacy. Besides, the paper is well written and easy to follow.

I have some concern for the paper.
(1) the novelty is fairly limited. Two major techniques:  graph contrastive pretraining and prompt tuning, have been studied before.

**Questions:**

My questions:
(1) SMART employed the GIN architecture as the graph encoder. I am wondering any other graph encoders have been tried compared to GIN.
(2) In the equation (6),  the layer K is a hyper parameter? It can impact the performance of SMART?
(3) For the mix-up strategy, SMART only chooses linear interpolation. Any reason for that? I am wondering if any other mix-up strategies have been tried compared to linear interpolation.

**Reviewer Confidence:**

3: The reviewer is confident but not certain that the evaluation is correct

**Scope:**

4: The work is relevant to the Web and to the track, and is of broad interest to the community

---

### Decision · Program_Chairs · 2024-01-22

**Decision:**

Accept

**Comment:**

This paper proposes a transfer-learning based approach for few-shot graph classification, called SMART, which uses graph mixup and prompt-tuning.

 Reviewers raised some concerns pre-rebuttal:
 - sensitivity -- to graph encoders (DT48), general hyperparameters (DT48, nwEq)
 - confusions about motivation and alignment with NLP (nwEq)
 - limited technical novelty (Q9kC, DT48)

 The authors responded in detail, and reviewers rated this paper generally favorably despite the above concerns. I encourage the authors to continue to revise their final draft with these considerations.